# Porous Mullite Ceramic Modification with Nano-WO_3_

**DOI:** 10.3390/ma16134631

**Published:** 2023-06-27

**Authors:** Ludmila Mahnicka-Goremikina, Ruta Svinka, Visvaldis Svinka, Vadims Goremikins, Svetlana Ilic, Liga Grase, Inna Juhnevica, Maris Rundans, Toms Valdemars Eiduks, Arturs Pludons

**Affiliations:** 1Institute of Materials and Surface Engineering, Faculty of Materials Science and Applied Chemistry, Riga Technical University, Paula Valdena st. 3/7, LV-1048 Riga, Latvia; ruta.svinka@rtu.lv (R.S.); visvaldis.svinka@rtu.lv (V.S.); liga.grase@rtu.lv (L.G.); inna.juhnevica@rtu.lv (I.J.); maris.rundans@rtu.lv (M.R.); toms-valdemars.eiduks@rtu.lv (T.V.E.); arturs.pludons@rtu.lv (A.P.); 2Institute of Structural Engineering and Reconstruction, Riga Technical University, Kipsalas st. 6A, LV-1048 Riga, Latvia; vadims.goremikins@rtu.lv; 3Department of Materials, Vinča Institute of Nuclear Sciences—National Institute of Republic of Serbia, University of Belgrade, Mike Petrovića Alasa 12-14, P.O. Box 522, 11001 Belgrade, Serbia; svetlanailic@vin.bg.ac.rs

**Keywords:** mullite, aluminum tungstate, zircon, porous ceramic, tungsten oxide, zirconia

## Abstract

Mullite and mullite–alumina ceramics materials with dominance of the mullite phase are used in different areas of technology and materials science. Porous mullite ceramics materials can be used simultaneously as refractory heat insulators and also as materials for constructional elements. The purpose of this work was to investigate the WO_3_ nanoparticle influence on the evolution of the aluminum tungstate and zircon crystalline phases in mullite ceramics due to stabilization effects caused by different microsize ZrO_2_ and WO_3_. The use of nano-WO_3_ prevented the dissociation of zircon in the ceramic samples with magnesia-stabilized zirconia (MSZ), increased porosity by approximately 60 ± 1%, increased the intensity of the aluminum tungstate phase, decreased bulk density by approximately 1.32 ± 0.01 g/cm^3^, and increased thermal shock resistance by ensuring a loss of less than 5% of the elastic modulus after 10 cycles of thermal shock.

## 1. Introduction

The relevance of new investigations in mullite ceramic synthesis and modification, as well as of new areas of mullite ceramics application, has been going on for 10 years. The popularity of research on mullite ceramics has increased significantly over the past 4 years (from 2019 to 2022); the increased number of published research articles confirms that. Moreover, mullite ceramics are actively being studied as high-temperature filters and thermal insulation materials [1,2]. Such thermal insulation mullite ceramic can be used to prevent heat losses and heat release to the environment, as well as to avoid damage caused by contact with high-temperature objects [2,3,4,5,6,7].

Improving the thermal insulation properties and thermal stability of mullite ceramics can be observed by increasing such material’s porosity, modifying by the addition of mullite whiskers [8,9] or by in situ formed mullite whiskers [10,11], or with nanosized oxides [12,13,14]. The pores’ morphology and size have important significance for refractory castables’ thermal properties. Ceramic structure, crystal morphology, and pore size can be changed by ceramic modification with nanoparticles [12,15]. The high specific surface area of nanometer size components increases the reactivity of particles and, as a result, the temperature required to start the sintering process will be decreased. The addition of nanosized oxides to refractory ceramics has an impact on the material’s properties. Research work has adopted nanosized TiO_2_, ZrO_2_, and MgO as sintering additives for technical ceramic modification [16]. Therefore, previous work has shown that nano-TiO_2_ could enhance the mechanical properties, for example, of high-alumina castables [17], yttria-stabilized zirconia (3YSZ) ceramics [18], and zirconia/alumina ceramics [19]. Esmaili et al. [20] reported that the addition of nano-TiO_2_ promotes mullitization, increases sintering, and improves the mechanical properties of porcelains. In the case of MgO–CaO refractory ceramic, the addition of nano-ZrO_2_ improves densification and resistance to hydration [21]. Gogtas et al. reported that nano-yttria-stabilized zirconia and nano-ZrO_2_ particles [22] improve the toughness of Al_2_O_3_-based matrices. There is growing interest in the use of oxides, such as WO_3_, to change the properties of technical ceramic. Due to such properties as catalytic activity [23], the existence of amorphous, mesosized, and nanosized porous structures, abundant quantity on earth, and nontoxic nature, WO_3_ has found wide application in the different fields of technical ceramics. Micro- and nanosize powder WO_3_ can be applied as ceramics for functional materials, capacitors, and varistors; special ceramic membranes for photocatalytic processes; ceramics for gas detection; special ceramic films; and thermoelectric ceramics for high-temperature applications [24,25,26,27]. Paul et al. investigated highly dispersed silver nanoparticles on tungsten oxide nanorods for photoelectrocatalytic CO_2_ reduction under visible light irradiation [28] and as effective catalysts of benzyl alcohol dehydrogenation [29]. The influence of WO_3_ on refractory porous mullite ceramic has been little studied. Our previous investigations confirmed that micro-WO_3_ application as a mullite doping additive accelerates the process of mullitization, which allows for decreasing the sintering temperature of mullite ceramic [30]. The scientific interest in research is caused by the lack of detailed studies on the influence of nano-WO_3_ on mullite ceramics’ functional properties.

The novelty of this study lies in the use of nano-WO_3_ in combination with micro-ZrO_2_ and micro-WO_3_ and the investigation of the mullitization improvement in porous mullite ceramic. An increase in thermal shock resistance and the formation of phases with a low or negative coefficient of linear thermal expansion allow for the use of such ceramics both at elevated and sharply changing temperatures.

## 2. Raw Materials and Testing Methods 

### 2.1. Raw Materials

Two different Nabalox aluminas—α-Al_2_O_3_ with an average particle size of 2 μm and γ-Al_2_O_3_ with d_50_ = 80 μm were purchased from Nabaltec AG of Germany. MEKA kaolin (d_50_ = 1.5 μm; SiO_2_—56.5 wt%, Al_2_O_3_—31.0 wt%) was supplied by the German company Amberger Kaolinwerke. Zirconia stabilized with 2.8 mol% of magnesia with an average particle size of 0.8 μm was purchased from Goodfellow, United Kingdom. Zirconia stabilized with 8 mol% of yttria with d_50_ = 0.5 μm, SiO_2_ amorphous with d_50_ = 3–5 μm, micro-WO_3_ with d_50_ = 5 μm, and nano-WO_3_ with d_50_ = 50 nm were supplied by GetNanoMaterials, France. Aluminum paste Aquapor-9008 with 70 ± 2% content of solid part and with d_50_ = 12 μm was acquired from German Schlenk Metallic Pigments GmbH. It is well known that a reduction in the raw component particle size promotes the sintering of ceramics due to increased contacting particle surface area. The purpose of choosing micro- and nanosized WO_3_ was to study the potential influence of the reduction in particle size, from micro- to nanosize, on the properties of the obtained ceramics.

### 2.2. Material Proportions

The base of the all mullite ceramics samples was prepared from 2 types of aluminas: alpha and gamma, kaolin, and amorphous SiO_2_. In order to meet mullite stoichiometry, Al_2_O_3_ and SiO_2_ were set at a 2.57:1 ratio. The total amount of Al_2_O_3_ consisted of ¾ γ-Al_2_O_3_ and ¼ α-Al_2_O_3_. Kaolin was used at 30 wt%. Different microsized ZrO_2_ sources, such as yttria-stabilized zirconia (8 mol% Y_2_O_3_—YSZ) and magnesia-stabilized zirconia (2.8 mol% MgO—MSZ), were used in a mixture with microsized WO_3_ for ceramic modifications. The ratios of the different stabilized microsized zirconia and microsized tungsten oxide were 1:1 and 1:2 of wt%. Nanosized WO_3_ was used at 1 wt% in all compositions. The A1 and A2 samples were the compositions with microsized yttria-stabilized zirconia and tungsten trioxide, respectively, in a 1:1 ratio plus nano-WO_3_ and 1:2 plus nano-WO_3_. The A3 and A4 samples were the compositions with microsized MSZ and microsized WO_3_, respectively, in a 1:1 ratio plus nano-WO_3_ and 1:2 plus nano-WO_3_.

### 2.3. Sample Preparation Method

Ceramics samples were prepared by slip casting of the concentrated raw materials slurry. The distilled water percentage was approximately 40 ± 2 wt%. A suspension of aluminum paste with water was added into the raw materials slurry for pore forming purposes and then mixed for approximately 10 min. Kaolin as a binder and uniform mixing using a laboratory mixer allows for obtaining a uniform distribution of all components in the final slurry and prevents sedimentation of particles. The prepared raw material slurries were slip casted into molds. The pores of the investigated mullite ceramics were obtained due to the H_2_ evolution that was formed due to the reaction between the Al paste and H_2_O. The samples were solidified at room temperature and dried at 100 °C for 24 h. The samples were synthesized at 1600 °C with a heating rate of 4.2 °C/min and with holding at T_max_ for 1 h. The fired samples’ cooling rate was the same as the rate of heating. The chosen heating rate was “industry average”—it was specifically chosen as a compromise between a slow heating rate and thus increased production costs and a fast heating rate, which, although being economically favorable, may be detrimental to product quality. 

### 2.4. X-ray Diffraction Analysis 

The crystalline phases of the sintered ceramics materials were analyzed using X-ray diffraction equipment BRUKER D8 Advance with a sealed ceramic copper X-ray tube, 40 kV voltage on copper anode, 40 mA of current intensity, detector 0d Sol-x, 0.1 mm of receiving slit, 0.6 mm of divergence slit, range of the measurement angle 10–80 2*θ*°, step scan mode, step size 0.008 2*θ*°, and time per step 6 s.

### 2.5. Scanning Electron Microscopy

The microstructures of the sintered samples were investigated using two scanning electron microscopes. The first was a Hitachi TM3000—TableTop SEM from Tokyo, Japan with an electron beam energy of 5 keV and 15 keV. The second high-resolution SEM was an FEI Nova NanoSEM 650 (Eindhoven, The Netherlands) with an electron beam energy of 10 keV. The low vacuum mode was used for microstructure observation; therefore, metal sputtering was not used. Energy dispersive mapping analysis was performed for certain samples. 

### 2.6. Apparent Porosity 

Archimedes’ principle and mathematical calculations were used for apparent porosity determination after soaking the samples in distilled water, respectively, based on European standard EN 623-2 [31]. The value of the apparent porosity *P* was calculated by Equation (1), and the value of water uptake *W* was calculated by Equation (2):
*P* = ((m_3_ − m_1_)/(m_3_ − m_2_)) × 100,(1)
*W* = ((m_3_ − m_1_)/m_1_) × 100,(2)
where m_1_ is the mass of the sample in the dry state, m_2_ is the apparent mass of the immersed sample, and m_3_ is the mass of the soaked sample.

### 2.7. Mercury Porosimetry

The distributions by pore size of the porous mullite ceramics were analyzed using mercury intrusion porosimetry with a US Quantachrome, Pore Master 33. Hg porosimetry allows for the determination of the pore size in the range of 0.1–1000 μm. The pore size determined by Hg porosimetry is related to the range of pressures to be used. The possible detectable pore diameter is inversely proportional to the pressure. Low-pressure mercury porosimetry determines macropores with pore diameters of 14–200 µm but in a vacuum at 360–950 µm. At first, mercury enters the larger pores, and then, as the total pressure slowly increases, it squeezes into the smaller pores. The pore shape affects the overall pore size distribution graph [32]. In practice, it is assumed that the pores have a cylindrical shape, which is characterized by the modified Jung–Laplace or Washburn Equation (3):(3)$$\Delta P=\frac{2\gamma \cos\theta }{r_{pora}}$$
where—pressure change when mercury enters the pore, N/m^2^, Pa;
*r_pore_*—corresponding pore radius, µm;*γ*—surface tension of mercury, N/m;*θ*—wetting angle between the solid surface of the material and mercury.

### 2.8. Thermal Shock Resistance Testing

The thermal shock resistance tests provided a measure of the ability of the ceramic materials to withstand several cycles of thermal stresses when subjected to rapid changes in temperature. Resistance to the rapid change in temperature by the scheme from 20 to 1000 °C with exposure for one hour at this temperature and with the subsequent sharp cooling at 20 °C was determined during 10 such cycles. 

### 2.9. E Modulus Measurement

The value of the E modulus was determined before and after the first, second, third, fifth, and tenth thermal shock tests using an acoustic impulse excitation method with Buzz-O-Sonic 5.0 equipment from BuzzMac International, LLC, Portland, ME, USA. This method is nondestructive and allows for the use of the same samples for all thermal shock cycles. The operating principle of the Buzz-O-Sonic system is based on the excitation of elastic oscillations with an impulse in the material; the created sound is perceived with a microphone connected to a computer. The acoustic system analyzes sound using the fast Fourier transform algorithm. This acoustic signal contains the sample’s natural frequencies of vibration, which are proportional to the elastic (Young’s) modulus, calculated with Buzz-O-Sonic 5.0 software according to classical beam theory [33]. 

## 3. Results and Discussion

### 3.1. Crystalline Phase Compositions

The X-ray diffraction analysis patterns of the sintered ceramics materials are displayed in Figure 1 and Figure 2. The use of micro- and nano-WO_3_ had an influence on the phase composition of the modified porous mullite ceramic. The main crystalline phase of the sintered samples corresponds to mullite (Figure 1). The phase of corundum is slightly observed for the A1 samples (Figure 1a). The mullite ceramic samples with YSZ (A1 and A2) contained monoclinic ZrO_2_ (Figure 1a,b). The samples with magnesia-stabilized zirconia (A3 and A4) additionally contained the zircon phase, but the intensity of monoclinic tetragonal ZrO_2_ greatly decreased (Figure 1b,c). The formation of zircon could be explained by the reaction of amorphous SiO_2_ and ZrO_2_ in an equimolar ratio at a temperature of 1200 °C [34,35]. Porous mullite ceramic A1 and A2 samples do not contain the zircon phase due to its dissociation with mullite phase formation after forming a liquid phase in the Y_2_O_3_–Al_2_O_3_–SiO_2_ system at 1550 °C [36,37]. The addition of nano-WO_3_ did not affect zircon formation in the cases of the A1 and A2 samples. The use of nano-WO_3_ in compositions with magnesia-stabilized zirconia and microsize WO_3_ in 1:1 and 1:2 ratios, respectively, caused the formation of a zircon phase due to preventing zircon dissipation in the MgO–Al_2_O_3_–SiO_2_ system in possibly emerging liquid at 1450–1550 °C, as in the case of the Y_2_O_3_–Al_2_O_3_–SiO_2_ system.

Figure 2 displays the X-ray diffraction patterns from 10° *2θ* to 35° *2θ* of the prepared ceramics. It can be seen that, as expected, all samples have crystalline as well as amorphous phases. The A1 ceramic samples from the composition of the YSZ and WO_3_ mixture in a 1:1 ratio and nano-WO_3_ have additional phases as orthorhombic and hexagonal WO_3_ (Figure 2a) [38,39,40]. The A2 samples have orthorhombic and monoclinic WO_3_ (Figure 2b). Figure 2c shows that in the case of the A3 samples, the diffraction peaks correspond to the planes of (002), (200), (020), (120), (022), and (202), proving the monoclinic phase of tungsten trioxide. It is also possible that a slight presence of the aluminum tungstate phase is presented in these samples, because some intensity peaks of Al_2_(WO_4_)_3_ and WO_3_ overlap each other. The XRD patterns of the A4 sample (Figure 2d) show the presence of Al_2_(WO_4_)_3_. The separate diffraction peaks in the 2*θ*° range from 14° to 26°, at 30.5°, 31.5°, 34°, and 35° correspond to the aluminum tungstate phase [41,42].

### 3.2. Macrostructure and Microstructure

Figure 3 shows the macrostructure and microstructure of the A1 samples. The A1 samples consist of densely packed mullite crystals. In turn, between the mullite crystals, there is a glassy phase containing WO_3_ that confirms the EDS point analysis in Table 1. Such a glassy phase is uniformly distributed at the structure and displays as lighter parts on the SEM micrographs in Figure 3a,b.

The structure of the porous mullite ceramic samples with a mixture of YSZ and tungsten trioxide in a 1:2 ratio and nano-WO_3_ additive (A2 sample, Figure 4) is more loose than the structure of the A1 sample. The glassy phase containing different WO_3_ is also shown as white particles and areas on the SEM micrographs.

Figure 5 shows the evenly distributed agglomerates of monoclinic WO_3_ in the structure, seen as white areas on the SEM of the A3 sample. The faceted zircon crystals can be visible at higher SEM magnification, as shown in Figure 5d.

SEM and mapping EDS analysis were performed for the porous mullite ceramic A4 samples, the raw material compositions of which contained a mixture of magnesia-stabilized zirconia and tungsten trioxide in a 1:2 ratio and nano-WO_3_ additive (Figure 6 and Figure 7). Figure 6a,b show the white phase that according to XRD analysis corresponds to aluminum tungstate. Its presence is clearly observed in many areas, as determined by EDS analysis (Figure 7). 

The SEM and mapping EDS analysis in Figure 7 allow for the designation of two types of mullite crystals that differ in size and aspect ratio. Two areas define the mullite crystals after the EDX results: the A and C phases (red and blue colors, respectively). The A4 samples’ structure is formed as thick long acicular crystals as well as thin elongated needle-shaped mullite crystals. The C phase corresponds to larger mullite crystals, but the A phase corresponds to thinner mullite crystals after EDX mapping results (Table 2). The width to length ratio of the thick mullite crystals is in the 15–22:1 interval. The needle-like thin mullite crystals are showing an aspect ratio of approximately 11:1 (with a length of 0.5–1.5 μm and a width of 0.1–0.2 μm).

Figure 7 shows that phase B (green color) takes a lower percentage place among the other phases. Phase B corresponds to the phase of zircon according to the elemental composition by EDX analysis of the A4 sample. The EDX mapping and Table 2 show that the D phase (yellow color) has a high weight percentage of such chemical elements as O, Al, Si, and W but a low weight percentage of zirconium, which can confirm the aluminum tungstate crystalline phase, as well as the aluminum tungstate glassy phase in these areas. In turn, the elemental composition of the A4 sample (Figure 8) shows that the W element is evenly distributed in the sample structure, in contrast to the distribution of the Zr element. By comparing the XRD results and EDX of the A4 sample, it can be concluded that the aluminum tungstate phase is evenly distributed in the structure of these samples.

The SEM results reveal that the morphologies of the microstructures of the A1 and A3 samples present significant similarities due to the amorphous phase between the mullite crystals. In turn, the amorphous phase is not observed between the mullite crystals in the A2 and A4 samples.

### 3.3. Bulk Density and Change in Samples’ Dimensions 

The bulk density and percent change in the samples’ dimensions after sintering or sample shrinkage are shown in Figure 9. The A1 sample that was modified with a mixture of yttria-stabilized zirconia and tungsten trioxide in a 1:1 ratio plus nano-WO_3_ has a higher bulk density of approximately 1.60 ± 0.05 g/cm^3^. The other samples, A2, A3, and A4, have a bulk density of approximately 1.25 ± 0.05 g/cm^3^. The A1 sample has a more densely packed structure due to the presence of an amorphous phase around the mullite crystals. The A1 and A2 samples from the compositions with YSZ and WO_3_, respectively, in a 1:1 ratio and 1:2 plus nano-WO_3_ have higher shrinkage than the A3 and A4 samples from the compositions with MSZ and WO_3_, respectively, in a 1:1 ratio and 1:2 plus nano-WO_3_. The usage of YSZ increased the shrinkage of the samples by ≈10% in comparison with the samples with MSZ due to the formation of a liquid phase in the Y_2_O_3_–Al_2_O_3_–SiO_2_ system.

### 3.4. Apparent Porosity, Water Uptake, and Pore Sizes 

The apparent porosity of the sintered samples is shown in Figure 10. The A1 sample with the higher bulk density has a smaller apparent porosity of approximately 40 ± 2% and water uptake of approximately 25 ± 1%. This is due to the YSZ (8 mol% Y_2_O_3_), which influenced the formation of a liquid phase that decreased the porosity. Doubling of WO_3_ prevents liquid formation in the system of Y_2_O_3_–WO_3_, as well as in samples with magnesia-stabilized zirconia. The samples of A2, A3, and A4 have porosity higher than 63 ± 1% and water uptake of approximately 40 ± 1%.

Figure 11 shows the pore size distribution after mercury intrusion porosimetry. One or two ranges of pore size distributions are observed in the investigated ceramics. In comparison with the other samples, the A1 sample has one narrow range of pore size distributions of approximately 2−15 μm, with pronouncing of pores of 8 μm size. It can be assumed that due to the composition, the formed amorphous phase partially filled pores larger than 15 microns, and thus only pores in a smaller size range remained. The other samples, A2, A3, and A4, have two ranges of pore size distributions. These ranges are from 2 μm to 15 μm and from 15 μm to 1000 μm. The most pronounced pore sizes in these samples are approximately 5−7 μm and 100−200 μm.

### 3.5. Resistance to Thermal Shock

The relative changes in the E modulus of the sintered samples after the thermal shock tests (10 cycles) are shown in Figure 12. All samples have high stability from the first to the fifth cycle of the rapid temperature change by the scheme from 20 °C to 1000 °C and back to 20 °C with holding for one hour at a temperature of 1000 °C. In this case, the elastic modulus losses do not exceed 5%, which is a high rate. The thermal shock resistance for each sample depends on individual factors. The A1 and A4 samples have a small relative change in the E modulus after the testing of 10 thermal shock cycles, which is less than 5%. In the case of the A1 sample, the positive influence lies in the presence of evenly distributed pores, with a range of pore size distributions from 2 μm to 15 μm and a phase composition of mullite and monoclinic ZrO_2_, which prevents the formation and growth of cracks.

The A2, A3, and A4 samples have two distinct arrays of pore sizes of relatively small (2–15 μm) and larger (15–1000 μm) size. The values of the E modulus of the A2 and A3 samples decreased significantly after the 10th cycle and, respectively, are 20% and 15% due to the preservation of WO_3_ with high LTEC and critical crack growth in the process of the 10th cycle. The A4 samples can better resist the sudden change in temperature due to the crystallinity structure and phase compositions from mullite, zircon, and aluminum tungstate. Mullite and zircon have low thermal expansion, respectively, from 4.0 to 5.9·10^−6^·°K^−1^ [43] and 4.1·10^−6^·°K^−1^ [44] between temperatures of 30 °C and 1000 °C, and aluminum tungstate has a negative thermal expansion of −1.5·10^−6^·°K^−1^ in the temperature diapason of 25−850 °C [45], which prevents thermal stress formation at the time of the thermal shock.

## 4. Conclusions

The influence of porous mullite ceramic modification with differently stabilized microsized ZrO_2_ and WO_3_, as well as with nanosized WO_3_, on the different properties of highly porous mullite ceramic materials was investigated. The use of kaolin as a binder, as well as the mixing of the initial raw materials with a technical mixer and raw material suspension slip casting, make it possible to obtain a uniform distribution of raw materials, including nanosized ones. In combination with these oxide compositions, the addition of a small percentage of nano-WO_3_ had an effect on the formation of aluminum tungstate with negative LTEC and zircon with low LTEC. The use of nano-WO_3_ prevented the dissociation of zircon in the ceramic samples with magnesia-stabilized zirconia and increased the intensity of the aluminum tungstate phase in the investigated materials with MSZ and WO_3_ in a 1:2 ratio, which made it possible to obtain mullite ceramics with increased resistance to thermal shock.

## Figures and Tables

**Figure 1 materials-16-04631-f001:**
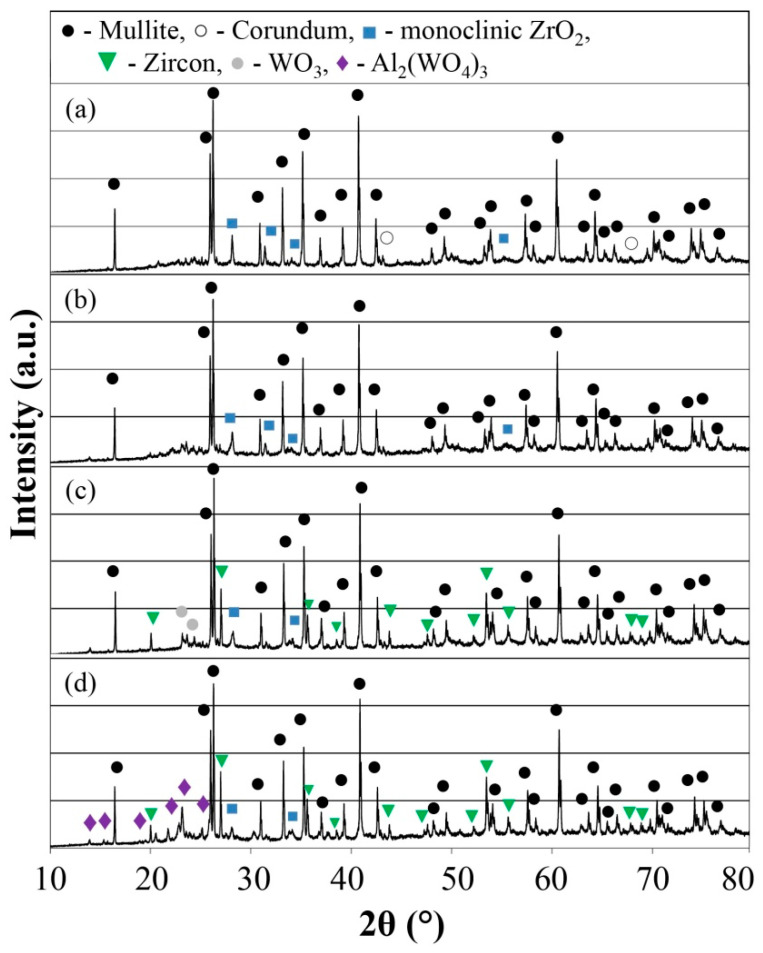
XRD of the investigated porous ceramic samples: (**a**) Sample A1; (**b**) Sample A2; (**c**) Sample A3; (**d**) Sample A4.

**Figure 2 materials-16-04631-f002:**
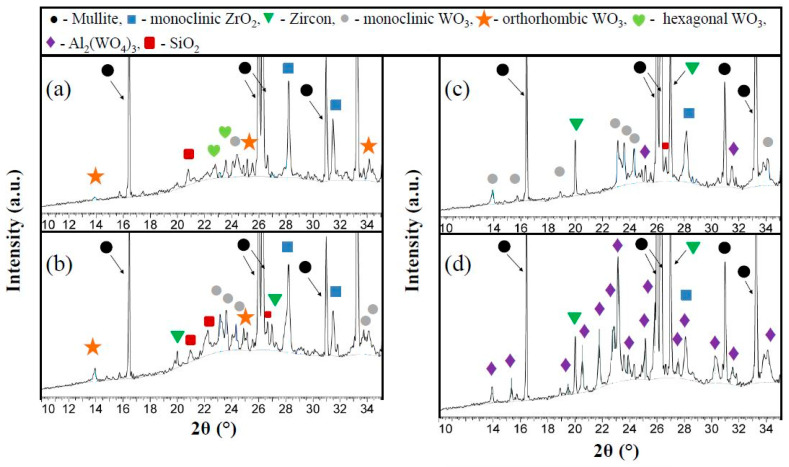
X-ray diffraction data from 0° 2*θ* to 35° 2*θ* of the sintered porous ceramic materials: (**a**) A1 sample; (**b**) A2 sample; (**c**) A3 sample; (**d**) A4 sample.

**Figure 3 materials-16-04631-f003:**
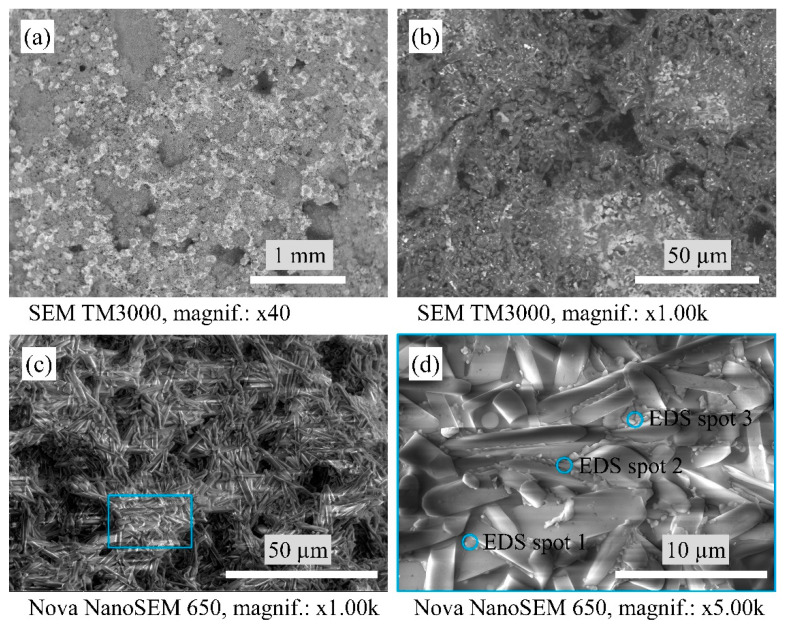
Micrographs of the sintered A1 samples’ microstructure with SEM.

**Figure 4 materials-16-04631-f004:**
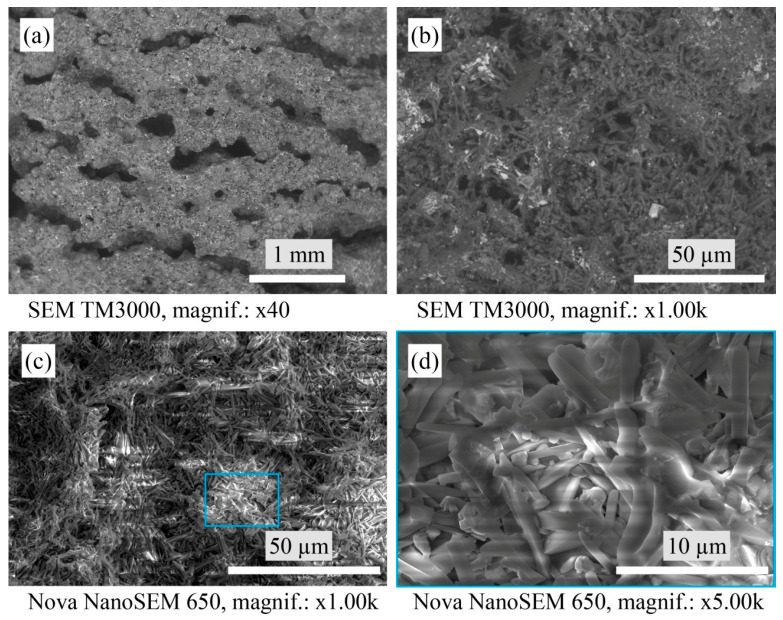
Micrographs of the sintered A2 samples’ microstructure with SEM.

**Figure 5 materials-16-04631-f005:**
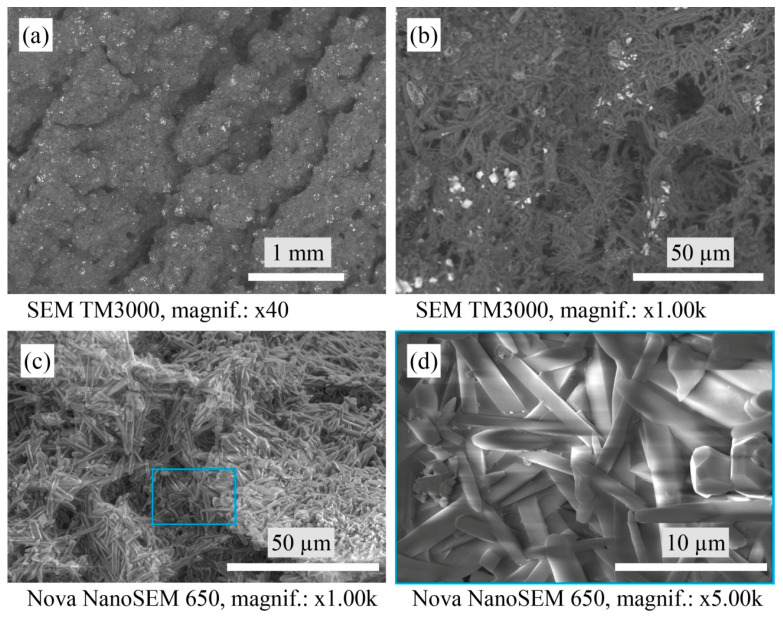
Micrographs of the sintered A3 samples’ microstructure with SEM.

**Figure 6 materials-16-04631-f006:**
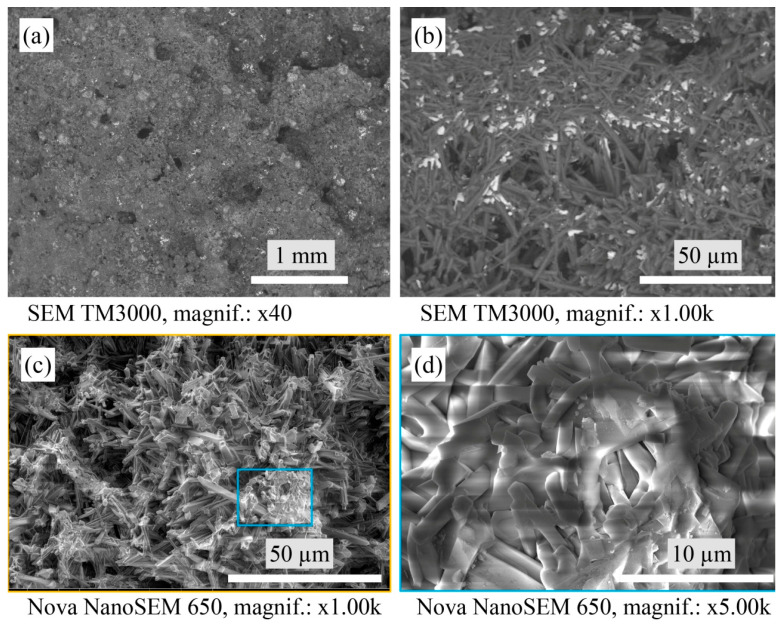
Micrographs of the sintered A4 samples’ microstructure with SEM.

**Figure 7 materials-16-04631-f007:**
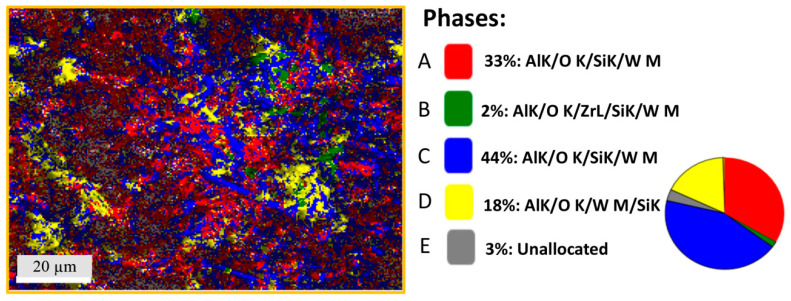
Energy dispersive spectroscopy mapping analysis of the sintered A4 sample. The EDX micrograph corresponds to the SEM micrograph in Figure 6c.

**Figure 8 materials-16-04631-f008:**
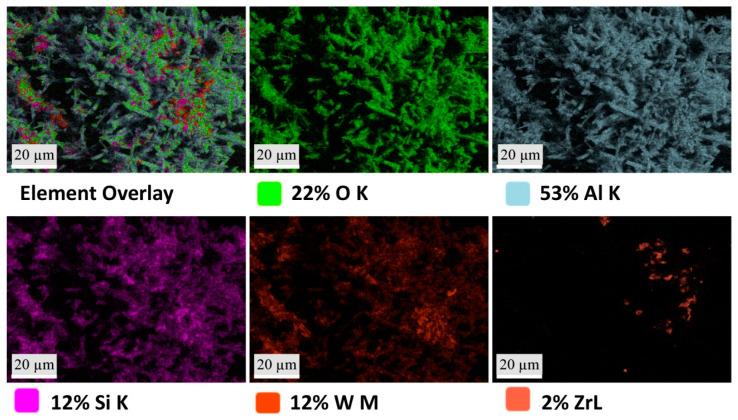
SEM–EDX micrographs of the element distribution in the structure of the A4 samples.

**Figure 9 materials-16-04631-f009:**
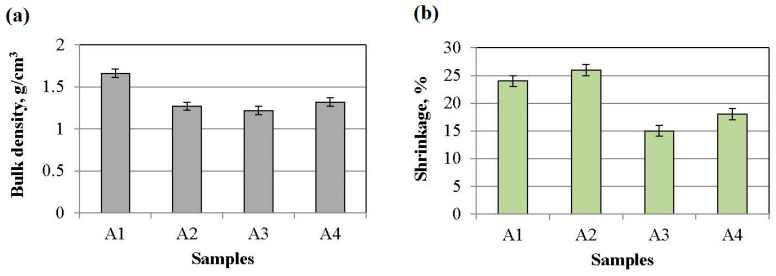
Sample characterizations: (**a**) bulk density and (**b**) shrinkage.

**Figure 10 materials-16-04631-f010:**
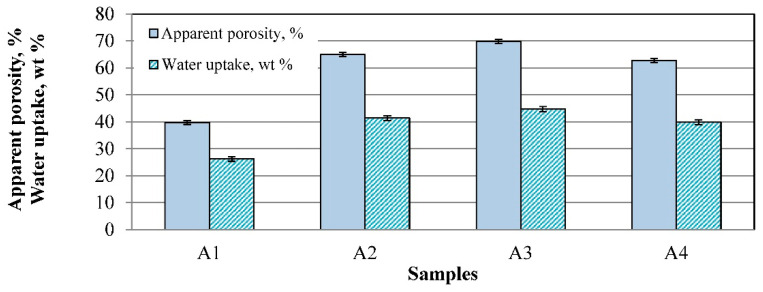
Apparent porosity of the investigated samples.

**Figure 11 materials-16-04631-f011:**
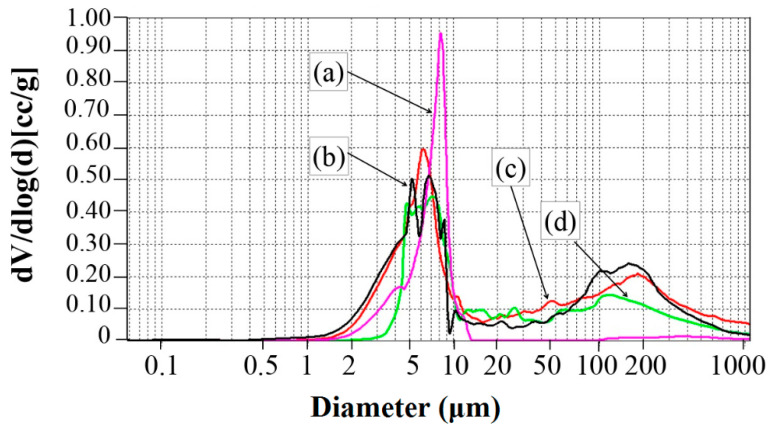
Distributions of the samples’ pore sizes: (a) sample A1; (b) sample A2; (c) sample A3; (d) sample A4.

**Figure 12 materials-16-04631-f012:**
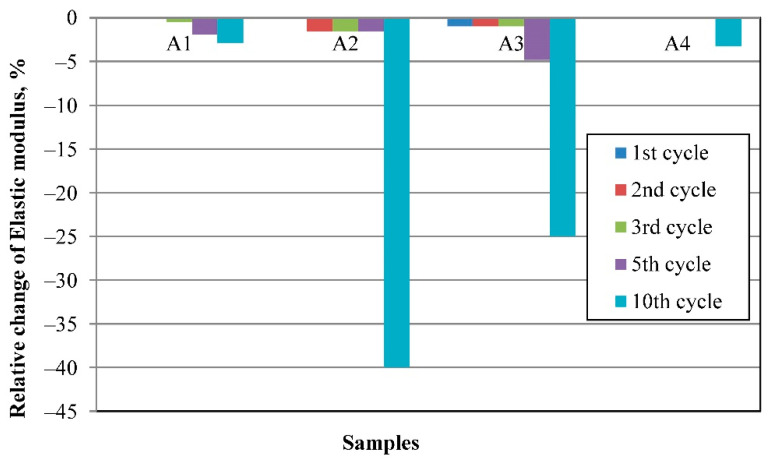
E modulus relative change after thermal shock testing of the samples.

**Table 1 materials-16-04631-t001:** Elemental composition with EDX point analysis of the A1 sample.

	EDS Spot 1		EDS Spot 2		EDS Spot 3	
Elements	wt%	Atomic%	wt%	Atomic%	wt%	Atomic%
O K	42.4	65.9	38.5	70.4	47.3	70.7
AlK	23.5	21.7	18.8	20.4	20.6	18.3
SiK	9.5	8.4	3.9	1.3	8.4	7.2
W M	19.7	2.7	34.0	5.4	19.4	2.5
Y L	3.3	0.9	3.3	1.1	2.7	0.7
ZrL	1.6	0.4	1.5	1.4	1.6	0.6

**Table 2 materials-16-04631-t002:** Elemental composition with EDX analysis of the A4 sample.

	Phase A		Phase B		Phase C		Phase D	
Elements	wt%	Atomic%	wt%	Atomic%	wt%	Atomic%	wt%	Atomic%
O K	33.6	46.9	43.3	67.9	48.5	62.8	44.0	61.9
AlK	49.0	40.5	13.0	12.2	34.8	26.7	24.6	20.4
SiK	15.6	12.4	13.7	12.2	13.6	10.0	20.3	16.2
W M	1.8	0.2	3.8	0.5	2.6	0.3	10.3	1.3
ZrL	0	0	26.2	7.2	0.5	0.2	0.8	0.2

## Data Availability

The data presented in this study are available on request from the corresponding author.

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
