# Peer review of "Porous Mullite Ceramic Modification with Nano-WO3"

_materials, 2023, doi:10.3390/ma16134631_

Round 1

Reviewer 1 Report

The authors successfully fabricated the porous mullite ceramic and investigated the influence of the porous mullite ceramic modification with differently stabilized micro sized ZrO2 and WO3 as well as with nano sized WO3 on the different properties of the highly porous mullite ceramic materials. The work can be accepted for publication in Materials after some issues below are addressed carefully.

 1. The abbreviation that appears for the first time must use the full name. For example: what does MSZ mean?

2. Latest literatures on mullite related ceramics should be mentioned in the introduction part. For example, Journal of Alloys and Compounds 893 (2022) 162231

3. About the catalytic activity of WO3, related literature should be mentioned. Journal of Advanced Ceramics, 2022, 11(8): 1208-1221

 4. Why are two types of alumina added and what are the standards for controlling their ratio? Has the author conducted a comparative experiment?

 5. Page 85. “The ratios of the different stabilized micro sized zirconia and micro sized tungsten oxide were 1:1 and 1:2” mol percent or wt percent?

6. Page 165. “It can be seen that all samples have crystalline, as well as amorphous phases.” All ceramics should have crystalline and amorphous phases.

7. page 181. “between the mullite crystals there is a glassy phase containing WO3” How to determine the composition of glass phase?

8. Figure 9. It seems the densities of green bodies are different? A2 has much low green density, compared with A1.

9. Why the apparent porosity and water uptake is different?

10. How about the strength of the obtained ceramics?

11. The article says that WO3 prevents the formation of the liquid phase, so how did it prevent it? The mechanism of its influence needs to be elaborated.

 12. Some characterization tests require analyzing the reasons and mechanisms, rather than simply describing them.

English writing must be checked.

Author Response

The authors are truly thankful for the constructive comments and recommendations made for the improvement of the first version of the paper. We took a great care to reflect the required changes and to address with as much details as possible the comments made that required further explanation.

Reviewer 2 Report

In this manuscript, Mahnicka-Goremikina  et. al has shown the use of porous mullite ceramic modification with nano-WO3 for refractory heat insulators and also as materials for  constructional elements. The results are interesting and the knowledge obtained from this study is helpful to such material studies. It can be only accepted for the publication in Materials.

Detailed comments are as follows:

1)      The manuscript does not present the novelty of the work clearly in the aspects of the functionalization and qualification/quantification, modification, properties and explaining the activity of the introduced method and new material. The introduction section needs to be revised.

2)      The author should eliminate the current grammatical, spelling, verb tense singular and plural, punctuation mark errors (commas, italics and so on) and also should confirm the correct scientific English. It is suggested to avoid or insert commas on the right positions. The English of the manuscript needs improvement. The authors should work and elevate the scientific English of the manuscript.

3)      The authors should mention the complete terms of abbreviations before the parenthesis for the first time use when an abbreviation is used on the abstract and main manuscript.

4)      The authors should revise the introduction section of the manuscript to clarify and justify the importance of the project. Otherwise, there is no advantage or novelty introduced on this project.

5)      The authors should explore/include the previously similar published articles and compare them with their strategy on the introduction of the revised manuscript (Journal of CO2 Utilization 41, 2020, 101284; Catalysis Communications 132, 2019, 105804; Materials Letters 210, 2018, 109-112).

6)      The abstract should be revised to clearly show the novelty of the work.

7)      The author should index the peaks in the XRD pattern. All the peaks in figure no. 2 is not indexed.

8)      The characterization of the materials is incomplete.

9)      The author should provide wide XPS spectrum and high resolution XPS spectra.

10)  The role of kaolin as a binder should be discussed more in details.

The author should eliminate the current grammatical, spelling, verb tense singular and plural, punctuation mark errors (commas, italics and so on) and also should confirm the correct scientific English. It is suggested to avoid or insert commas on the right positions. The English of the manuscript needs improvement. The authors should work and elevate the scientific English of the manuscript.

Author Response

(The authors gave the same response as above.)

Reviewer 3 Report

Dear authors, the work is well presented, and it is acceptable after a few reversions.

1. the tile indicates that Nano-WO3 was used, but in the experimental section, it seems micro-WO3 was used. Please confirm it and give the actual size of WO3 in the study.

2. Please check through the manuscript and ensure that all the subscript is correct.

3. I suggest the parameter used in the figures are unified in the same manner such as diameter, μm is diameter (μm).

The English is fine.

Author Response

(The authors gave the same response as above.)

Reviewer 4 Report

The presented manuscript contain interest experimental data and discussion.

The manuscript need to improved.

1. In the lines 39, 49, 59 and 155 missed the references [Error!]

2. Section 2.1. What is the reasons for choosing the sizes of used particles? 

3. Section 2.3. What is the reason for choosing the heating rate of 4.2 C/min? 

4. Figure 1. Authors need to calculate the unit cell parameters with Rwp parameters for all phases and debated the results in comparison to unit cell parameters from JCPDS database.

5. Line 246. Authors need to correct typo. 1.25+/-0.05.

6. Figure 9. The quality of Fig.9 need to improved.

Minor editing of English language required

Author Response

(The authors gave the same response as above.)

Round 2

Reviewer 3 Report

after the careful reversion made by the authors, the manuscript can be accepted in its current form.